# Antimetastatic Activity of Apoptolidin A by Upregulation of N-Myc Downstream-Regulated Gene 1 Expression in Human Colorectal Cancer Cells

**DOI:** 10.3390/ph16040491

**Published:** 2023-03-26

**Authors:** Kay Zin Kyaw, Jiyoon Park, Seung Ho Oh, Ji Yun Lee, Eun Seo Bae, Hyen Joo Park, Dong-Chan Oh, Sang Kook Lee

**Affiliations:** College of Pharmacy, Natural Products Research Institute, Seoul National University, Seoul 08826, Republic of Korea

**Keywords:** colorectal cancer, NDRG1, apoptolidin A, G_0_/G_1_ cell cycle arrest, apoptosis, EMT, metastasis

## Abstract

Colorectal cancer (CRC) is one of the most prevalent tumors with high metastatic potential; consequently, finding new drug candidates that suppress tumor metastasis is essential. Apoptolidin A is a macrocyclic lactone produced by *Amycolatopsis* sp. DW02G. It exhibits significant cytotoxicity against several cancer cell lines, but its effects on CRC cells remain unknown. Therefore, the present study investigated the antiproliferative and antimetastatic activities of apoptolidin A and its underlying molecular mechanisms in CRC cells. Apoptolidin A effectively inhibited CRC cell growth and colony formation. The induction of G_0_/G_1_ phase cell cycle arrest was associated with the downregulation of cyclin D1 and CDK4/6 expression. Long-term exposure to apoptolidin A also induced apoptosis as confirmed by the downregulation and upregulation of Bcl-2 and Bax expression, respectively. Moreover, apoptolidin A effectively upregulated the suppressed expression of N-Myc downstream-regulated gene 1 (NDRG1), a tumor suppressor gene, in a concentration-dependent manner in CRC cells. The antimetastatic potential of apoptolidin A was also correlated with the expression of epithelial–mesenchymal transition (EMT) biomarkers, including the upregulation of E-cadherin and downregulation of N-cadherin, vimentin, snail, and MMP9 in CRC cells. These findings suggest that apoptolidin A exerts antiproliferative and antimetastatic activities by regulating the NDRG1-activated EMT pathway in CRC cells.

## 1. Introduction

Colorectal cancer (CRC) is one of the major causes of cancer death worldwide [1]. It is an aggressive tumor, which causes distant metastasis. Approximately 20% of patients with CRC are first diagnosed with metastasis. Patients with metastasis also have a lower 5-year survival rate than those without metastasis due to there being fewer treatment options; consequently, more than 90% of CRC mortality is connected to distant metastasis [2,3,4]. Currently, surgical resection, adjuvant chemotherapy, and neoadjuvant chemotherapy are the main treatment options for CRC metastasis. However, surgery is only suitable in 10–20% of cases, and the 5-year survival rate of CRC metastasis is 15%, due to factors such as tumor location and size, unresectable cancer, patients’ comorbidities, and the presence of extrahepatic disease [5]. To date, chemotherapy is the primary approach for treating metastasis in clinical settings. 5-Fluorouracil was a typical chemotherapeutic medication before the 1990s. Oxaliplatin, capecitabine, and irinotecan were then introduced after the 1990s [6]. However, treatment for CRC remains challenging due to aggressive local invasion and metastasis [7]. Consequently, the inhibition of cancer cell migration, invasion, and adhesion may be a useful strategy for preventing CRC metastasis, and new drug candidates with potential antimetastatic activity are urgently needed.

Metastasis involves epithelial–mesenchymal transition (EMT), migration and invasion, intravasation, circulation and adhesion, extravasation, and colonization of cancer cells [8]. Accumulating evidence suggests that EMT is essential for metastasis in cancer cells including CRC [9,10,11]. EMT can also promote cancer cell invasion and migration by making cells lose their epithelial characteristics [12,13,14]. Therefore, the inhibition of the EMT process is a potential approach for preventing cancer cell metastasis [15]. In addition, matrix metalloproteinases (MMPs) are essential in the proteolysis of extracellular matrix (ECM) proteins for enhancing cancer cell invasion and metastasis. Among MMPs, MMP9 is strongly associated with cell invasion and metastasis [16,17]. Moreover, the overexpression of N-Myc downstream-regulated gene 1 (NDRG1) may inhibit invasion by suppressing MMP9 expression [18].

NDRG1 is an important multifunctional protein that participates in EMT [19,20]. NDRG1 downregulation can promote invasion in vitro, and NDRG1 expression in tumor cells plays an important role in regulating invasion [21]. Studies also reported that NDRG1 is crucial for controlling cancer cell metastasis [22,23,24,25] and that NDRG1 overexpression is associated with a lower metastasis rate and an increase in 5-year survival in the prostate, breast, cervical, ovarian, and CRC [26,27,28,29]. Clinical evidence also suggests that NDRG1 expression levels provide important prognostic information in patients with CRC [30,31]. Since NDRG1 overexpression may play an important role in inhibiting cancer cell invasion and metastasis, the procurement of the NDRG1 inducer by small molecules may be a promising strategy in the development of anticancer agents.

Apoptolidin A, a macrocyclic lactone, was previously isolated from *Nocardiopsis* sp., and our group recently isolated apoptolidin A from *Amycolatopsis* sp. DW02G, a bacterial strain discovered in the gut of *Dolichovespula media media* at Gwanak Mountain, the Republic of Korea [32,33]. Apoptolidin A exhibited significant cytotoxicity against several cancer cell lines [33]. Additionally, it suppressed the viability of lung cancer and glioblastoma cells by triggering the AMP-activated protein kinase stress response pathway [34]. Apoptolidin A was also recognized for its capacity to selectively trigger apoptosis in transformed cells [32]. The antiproliferative activity of apoptolidin A in cancer cells was also reported with the inhibition of mitochondrial F0F1-ATPase [35]. However, the antiproliferative and antimetastatic effects of apoptolidin A in CRC remain to be fully elucidated.

Therefore, this study aimed to investigate the effects of apoptolidin A on CRC cell proliferation and metastasis and its underlying mechanisms.

## 2. Results

### 2.1. Growth Inhibition Activity of Apoptolidin A in Human Colorectal Cancer (CRC) Cells

Apoptolidin A (Figure 1A) was reported to exhibit considerable cytotoxicity against various cancer cell lines [33]. However, its antiproliferative activity in CRC cells remains unknown. Therefore, the growth inhibition activity of apoptolidin A was evaluated in CRC cells using the sulforhodamine B assay. After 72 h of treatment with apoptolidin A, it effectively inhibited the growth of RKO, HCT116, and SW480 cells in a concentration-dependent manner. In addition, the IC_50_ value of apoptolidin A in a normal epithelial colon cell line (CCD841 CoN cells) was over 10 μM, indicating the relative selectivity against cancer cells compared to normal colorectal cells (Figure 1B, Table 1). RKO and HCT116 cell lines were more sensitive to apoptolidin A; thus, they were selected for further investigation. Apoptolidin A also inhibited colony formation in RKO and HCT116 cells (Figure 1C). These data suggest that apoptolidin A can markedly inhibit CRC cell proliferation and colony formation.

### 2.2. Effect of Apoptolidin A on Induction of Cell Cycle Arrest in CRC Cells 

Cell cycle arrest and induction of apoptosis are two fundamental mechanisms contributing to the inhibition of cell proliferation [36]. To determine the effect of apoptolidin A on the cell cycle phases of RKO and HCT116 cells, the cell cycle distribution was investigated using flow cytometry after staining with propidium iodide (PI). The G_0_/G_1_ cell populations were increased in RKO and HCT116 cells in a concentration-dependent manner (Figure 2A). The cell proportions in the G_0_/G_1_ phase were increased from 49.14% (vehicle-treated control) to 57.39% (4 μM) in RKO cells and from 52.91% (vehicle-treated control) to 61.56% (10 μM) in HCT116 cells. The deregulation of the cell cycle is one of the main characteristics of cancer cells. Cancer-related cell cycle abnormalities are caused by the upregulation of critical checkpoint genes, particularly the cyclin-dependence kinase (CDK) family [37]. To further identify the underlying mechanisms of cell cycle arrest, the expression of cell cycle checkpoint proteins was evaluated by Western blot. Figure 2B shows that the protein levels of cyclin D1, CDK4, and CDK6 were downregulated in both cell lines, indicating that the induction of G_0_/G_1_ cell cycle arrest by apoptolidin A is, in part, associated with the modulation of cell cycle checkpoint expression in CRC cells.

### 2.3. Effect of Apoptolidin A on Induction of Apoptosis in CRC Cells

Apoptolidin A was reported to induce apoptosis along with a mitochondrial F0F1-ATPase inhibitor [32,35]. To further determine whether apoptolidin A can induce apoptosis with long-term exposure, the cells were treated with apoptolidin A for 48 h, and then the DNA contents were measured by flow cytometry after labeling with annexin V-fluorescein isothiocyanate (V-FITC) and PI. The percentage of apoptotic cells (early and late apoptosis) increased from 8.25% (vehicle-treated control) to 26.35% (4 μM) in RKO cells and from 6.97% (vehicle-treated control) to 30.73% (10 μM) in HCT116 cells in a concentration-dependent manner (Figure 3A). The Bcl-2 protein family plays a crucial role in controlling the mitochondrial-dependent apoptotic process. The main proteins in the Bcl-2 family include Bcl-2, an antiapoptotic protein, and Bax, a proapoptotic protein [38]. Immunoblotting analysis was performed to investigate the mechanisms of apoptosis induction by apoptolidin A. When RKO and HCT116 cells were treated with the indicated concentrations of apoptolidin A for 48 h, the expression of Bax protein was upregulated, whereas that of Bcl-2 protein was downregulated (Figure 3B). These results showed that the induction of apoptosis by apoptolidin A is, in part, associated with mitochondrial-based apoptosis regulation.

### 2.4. Effect of Apoptolidin A on Regulation of NDRG1 Expression in CRC Cells

The expression of NDRG1, a metastasis suppressor, is markedly decreased in a number of malignancies, including CRC [39]. To confirm the expression levels of NDRG1, the protein expression of NDRG1 in a normal epithelial colon cell line (CCD841 CoN cells) and CRC cell lines (RKO, HCT116, and SW480) was assessed by immunoblot analysis. Figure 4A shows that the protein expression level of NDRG1 was relatively lower in CRC cells than in normal epithelial colon cells, indicating a negative correlation between NDRG1 expression and malignancy. To determine the effect of apoptolidin A on NDRG1 expression in RKO and HCT116 cells, the cells were treated with apoptolidin A for 24 h and examined via immunoblot and real-time polymerase chain reaction (PCR) analysis. In both cell lines, apoptolidin A increased NDRG1 protein and mRNA expression in a concentration-dependent manner (Figure 4B,C). These data indicate that the growth inhibition of CRC cells by apoptolidin A may be partly correlated with the upregulation of NDRG1 expression in CRC cells.

### 2.5. Effect of Apoptolidin A on Migration of CRC Cells

NDRG1 expression suppresses metastasis and, thus, may affect CRC prognosis [23]. Migration and invasion are essential processes for tumor cell metastasis [40]. Therefore, the effects of apoptolidin A on cell migration were investigated using wound healing assays. The findings revealed that apoptolidin A significantly inhibited cell migration in RKO (Figure 5A) and HCT116 cells (Figure 5B) in a concentration-dependent manner. Cancer cells migrate, invade, and adhere following EMT progression, demonstrating the importance of EMT in cancer cell metastasis [41]. EMT is characterized by the upregulation of mesenchymal biomarkers such as N-cadherin and vimentin and the downregulation of epithelial biomarkers including E-cadherin expression. EMT is also affected by transcription factors including snail [42]. To elucidate the underlying mechanism behind the antimigration activity of apoptolidin A, immunoblot analysis was used to investigate the effects of apoptolidin A on EMT-associated protein expression. Following 24 h of apoptolidin A treatment in RKO and HCT116 cells, the protein expression levels of the epithelial marker E-cadherin were increased, whereas those of mesenchymal markers N-cadherin, vimentin, snail, and MMP9 were decreased (Figure 5C). These findings suggest that the NDRG1-mediated EMT process may be, in part, involved in the antimigration activity of apoptolidin A in human CRC cells.

### 2.6. Effects of siNDRG1 on Cell Migration in Apoptolidin A-Treated CRC Cells 

To determine the significance of NDRG1 in apoptolidin A-treated RKO cells, a set of siRNAs was used to knock down NDRG1 expression in RKO cells. Figure 6 shows that the mRNA (Figure 6A) and protein expression (Figure 6B) of NDRG1 were markedly suppressed after selected siRNA transfection for 48 h. Based on the protein expression of silencing siRNA efficiency, siNDRG1-2 was selected for further investigation. To evaluate the role of NDRG1 in the antimigration activity of apoptolidin A-treated CRC cells, a wound healing assay was performed using apoptolidin A-treated NDRG1 siRNA transfected RKO cells. The data demonstrate that the antimigration activity of apoptolidin A was decreased in siNDRG1 transfected RKO cells compared with siRNA control transfected RKO cells (Figure 6C). These findings suggest that the antimetastatic activity of apoptolidin A was diminished by siNDRG1 co-treatment. These data confirm that the induction of NDRG1 by apoptolidin A was suppressed by siNDRG1, which may be, in part, associated with the regulation of NDRG1 by apoptolidin A in CRC cell metastasis. Western blot analysis also indicated that silencing NDRG1 in RKO cells reversed the regulatory effects of apoptolidin A on EMT-associated biomarkers (Figure 6D).

## 3. Discussion

Metastasis is one of the most critical and challenging problems in cancer treatment [43]. Among different types of tumors, CRC has significant metastatic potential, and its mortality is continuously increasing. Chemotherapy is the second most prevalent treatment option for CRC after surgery; however, it is limited by drug resistance and its side effects [44]. Consequently, more efficient strategies and new chemotherapeutic targets are urgently needed for CRC treatment. This study was designed to investigate the antiproliferative and antimetastatic activities of apoptolidin A in CRC cells.

A previous study reported that apoptolidin A inhibits the proliferation of several cancer cell lines [33]. The present study demonstrated the antiproliferative activity of apoptolidin A in the human CRC cell lines, RKO and HCT116. Therein, apoptolidin A significantly inhibited the proliferation and colony formation of RKO and HCT1116 cells. Restricted cell cycle progression and apoptosis evasion are common events in the development of colon cancer [45]. This study assessed whether apoptolidin A treatment changes cell cycle regulation in RKO and HCT116 cells. Apoptolidin A increases the proportion of cells in the G_0_/G_1_ phase, which results in the arrest of the G_0_/G_1_ phase cell cycle. Several cyclin–CDK complexes phosphorylate the proteins that attempt to control the cell cycle. While cyclin E and CDK2 are essential for the late stages of the G_0_/G_1_ cell phase, CDK4 and CDK6 are necessary for the development of the G_1_ phase through the formation of CDK4/6–cyclin D1 complexes [37]. In the present study, apoptolidin A suppressed the expression of cyclin D1, CDK4, and CDK6 proteins in a concentration-dependent manner, suggesting that apoptolidin A induces G_0_/G_1_ phase cell cycle arrest by modulating the expression of cell cycle checkpoint proteins.

Most anticancer drugs exhibit cytotoxicity against cancer cells by inducing apoptosis [46,47]. The Bcl-2 protein family plays a crucial role in controlling the mitochondrial-dependent apoptosis process. These intracellular proteins regulate pro- and antiapoptotic signals and the mitochondrial membrane potential. Bax triggers apoptosis by decreasing mitochondrial permeability in response to cellular stresses. On the other hand, Bcl-2 prevents cell death by inhibiting Bax [48,49,50]. A previous study reported that apoptolidin A promoted apoptosis by activating mitochondrial proteins [32]. The present study found that apoptolidin A induces apoptosis by increasing Bax protein expression but decreasing Bcl-2 protein expression in CRC cells. These findings are consistent with those of a previous report and suggest that apoptolidin A induces apoptosis through the mitochondrial apoptotic pathway.

In human breast, prostate, and CRC, NDRG1 was identified as a metastasis suppressor gene, suppressing cell growth and metastasis in vitro and in vivo [31,51,52]. Cancer cells that detach from the primary tumor migrate to distant organs and become secondary tumors. EMT progression induces cancer cells to lose cell-to-cell connections by adopting the mesenchymal characteristics necessary for cancer cell detachment. Therefore, the suppression of EMT is essential for regulating metastasis [53]. Detachable cancer cells migrate and invade the ECM, which is degraded by MMPs. MMPs are an ECM component that plays a critical function in tumor EMT. Therefore, inhibiting migration and invasion by reducing the expression and activity of MMPs might be an effective way to prevent metastasis [54,55,56]. Notably, NDRG1 has been reported to suppress EMT [20]. In addition, NDRG1 overexpression may reduce MMP9, thereby restricting cellular motility and invasion capability [57]. The previous study reported that overexpression of NDRG1 induced G_0_/G_1_ cell cycle arrest and apoptosis. Moreover, upregulation of NDRG1 inhibited cell migration of CRC and highly metastasis CRC cells [58]. In the present study, apoptolidin A significantly enhanced NDRG protein and gene expression. Moreover, the inhibition of migration activity by apoptolidin A was correlated with enhanced E-cadherin expression and decreased N-cadherin, vimentin, snail, and MMP9 expression in CRC cells. These results suggest that apoptolidin A exhibits antimigration activity by suppressing the expression of EMT-associated biomarkers in CRC cells.

NDRG1 has been suggested to be a tumor suppressor gene in numerous malignancies, and NDRG1 overexpression is negatively correlated with CRC metastasis [51]. In this study, apoptolidin A enhanced NDRG1 expression and inhibited EMT in CRC cells. Additional data revealed that silencing NDRG1 decreased the antimigration activity of apoptolidin A in RKO cells. Moreover, siNDRG1 treatment reversed the effect of apoptolidin A on EMT inhibition. Taken together, these findings indicate that apoptolidin A may exhibit antimetastatic effects on CRC cells via NDRG1-activated EMT inhibition. 

## 4. Materials and Methods

### 4.1. General Reagents 

Cultured media were purchased from HyClone Laboratories, Inc. (South Logan, UT, USA). All chemicals and reagents were obtained from Sigma-Aldrich (St. Louis, MO, USA, unless otherwise specified. Apoptolidin A (>98% purity by HPLC analysis) was isolated, as previously described, from *Amycolatopsis* sp. DW02G, a bacterial strain isolated from the gut of *Dolichovespula media media* collected from Gwanak Mountain in the Republic of Korea [33]. 

### 4.2. Cell Culture

Normal epithelial colon cell lines (CCD 841 CoN) and human CRC cell lines (RKO, HCT116, and SW480) were obtained from the American Type Culture Collection (Manassas, VA, USA). Normal epithelial colon cells were maintained in EMEM medium, and DMEM medium for RKO, SW480, and RPMI medium for HCT116 cells were used for cell culture. All cultured media were supplemented with 10% FBS, 10,000 units/mL penicillin G, and 10,000 μg/mL streptomycin. All cell lines were maintained at 37 °C in a humidified 5% CO_2_ atmosphere.

### 4.3. Cell Proliferation Assay

The proliferation of cells after compound treatment was measured using SRB assay [59]. Cells were seeded in 96-well plates with the test compound for 72 h. The cells were then fixed with 10% trichloroacetic acid (TCA) for 30 min at 4 °C, washed, and dried. After staining with 0.4% (*w*/*v*) SRB in a 1% (*v*/*v*) acetic acid solution for 2 h at room temperature, the plates were washed to remove unbound dye. After staining, 10 mM Tris (pH 10.0) was used to dissolve labeled cells and the absorbance was measured at 515 nM using a VersaMax ELISA microplate reader (Molecular Devices, Sunnyvale, CA, USA). The IC_50_ value was calculated by the following formula: cell survival rate (%) = (a − c)/(b − c) × 100 (a = absorbance at each concentration of the test compound, b = absorbance at vehicle-treated control, and c = absorbance of the 0-day seeding cells).

### 4.4. Gene Silencing by siRNA Transfection

siRNAs were purchased from Invitrogen (Invitrogen, Carlsbad, CA, USA). For control, negative control siRNA was also purchased from the same company. Following seeding, cells were transfected with a siRNA duplex for 48 h using Lipofectamine RNAimax in accordance with the manufacturer’s instructions (Invitrogen, Carlsbad, CA, USA). The coding strand for NDRG1 was as follows: NDRG1-1; sense CUU UUU GGG AAG GAA GAA A and antisense UUU CUU CCU UCC CAA AAA G, NDRG1-2; sense GAU ACA UGA GCC AGU GAU U and antisense AAU CAC UGG CUC AUG UAC C, NDRG1-3; sense GAG UAC GGA UGG GAA ACU A and antisense UAG UUU CCC AUC CGU ACU C.

### 4.5. Colony Formation

Cells (200 per well) were seeded into 6 well plates and incubated at 37 °C. On the next day, cells were treated with the indicated concentrations of the compound. Every 3 days, the culture medium was replaced with a fresh one. After 10–12 days, the cells were rinsed with PBS, fixed with 4% paraformaldehyde, and stained with 0.1% crystal violet. 

### 4.6. Immunoblotting

The compound-treated cells were lysed in a 2× sample loading buffer (Bio-Rad, Hercules, CA, USA). Protein concentrations were quantified by the bicinchoninic acid assay (BCA) method [60]. On SDS-PAGE gel, proteins were separated and transferred to polyvinylidene fluoride membranes (Millipore, Bedford, MA, USA). After blocking with 5% bovine serum albumin (BSA) in Tris-buffered saline containing 0.1% Tween-20 (TBST) for 1 h shaking, the membrane was probed with antibodies for anti-NDRG1 (#9485; 1:1000), anti-cyclin D1 (#55506; 1:1000), anti-CDK4 (#12790; 1:1000), anti-CDK6 (#13331; 1:1000), anti-Bcl-2 (#15071; 1:1000), anti-E-cadherin (#14472; 1:1000), anti-vimentin (#5741; 1:1000), anti-Snail (#3879; 1:1000), anti-MMP9 (#13667; 1:1000) (Cell Signaling Technology, Beverly, MA, USA), anti-beta-actin (#47778; 1:10,000), anti-Bax (#7480; 1:500) (Santa Cruz Biotechnology, Dallas, TX, USA), and anti-N-cadherin (#610920; 1:1000) (BD Biosciences, San Jose, CA, USA). Specific proteins were detected by secondary antibodies, and the chemiluminescence signals were detected using a Miracle-StarTM Western Blot Detection System (iNtRON Biotechnology, Seongnam, Republic of Korea). Image J (National Institutes of Health, Bethesda, MA, USA) was used for the quantitative analysis.

### 4.7. Cell Cycle Analysis 

The cells were seeded on plates, and, on the following day, cells were starved with a serum-free medium overnight. Cells were treated with the test compound for 24 h and collected. The cells were then washed twice with PBS, fixed in 70% cold ethanol, and incubated at −20 °C for 4 h or overnight. The cells were washed with cold PBS, and the fixed cells were resuspended with RNase A (100 g/mL) and shaken for 30 min. The cells were stained with PI (50 g/mL) for 15 min at room temperature, and the cell cycle analysis was conducted using flow cytometry (FACSCalibur flow cytometer; BD Biosciences, San Jose, CA, USA). The fluorescence intensity was analyzed using the program CellQuest 6.0 (BD Biosciences, San Jose, CA, USA).

### 4.8. Apoptosis Analysis with Annexin V-Fluorescein Isothiocyanate (V-FITC)/PI Double Staining by Flow Cytometry

The cells were seeded and treated with the test compound. The cells were then collected and washed with cold PBS. After being resuspended with 1× binding buffer, the cells were stained with annexin V-FITC and PI for 15 min. After dilution in a 1× binding buffer, the cells were immediately assessed using a flow cytometer (FACSCalibur flow cytometer; BD Biosciences, San Jose, CA, USA). 

### 4.9. Wound Healing Assay

Cells were seeded into a six-well plate. After incubating for 24 h (80–90% confluence), the cells were gently scratched to create a mechanical wound using Scratcher (SPL Life Sciences, Pocheon, Republic of Korea). The detached cells were removed using PBS and treated with the compound in a serum-free medium. Using a phase-contrast microscope, images were captured at 0 and 24 h and quantified in Image J. 

### 4.10. RNA Extraction and Real-Time PCR

RNA was isolated using the Trizol reagent (Invitrogen, Carlsbad, CA, USA). ReverTra Ace qPCR RT Master Mix (Toyobo, Osaka, Japan) was used to synthesize single-stranded cDNA from 1000 ng of extracted RNA. Real-time PCR was performed using iQ SYBR Green Supermix (Bio-Rad, Hercules, CA, USA) in accordance with the manufacturer’s instructions. The relative expression, normalized by β-actin, was calculated by comparative CT. The sequences of the primers used are the following: NDRG1: 5′-AGATCTCAGGATGGACCCAAG-3′ and 5′-TTGATGAACAGGTGCAGGTTG-3′ β-Actin: 5′-AGCACAATGAAGATCAAGAT-3′ and 5′-TGTAACGCAACTAAGTCAT A-3′ (Bioneer, Daejeon, Republic of Korea).

### 4.11. Statistical Analysis 

Student’s *t*-test or one-way ANOVA along with Dunnett’s *t*-test were used to evaluate statistical analyses. After calculating the means and standard deviations for at least three experiments, statistical significance was determined to be (* *p* < 0.05, ** *p* < 0.01, and *** *p* < 0.001).

## 5. Conclusions

In conclusion, apoptolidin A induces NDRG1 expression in RKO and HCT116 cells. The antiproliferative activities of apoptolidin A were confirmed by the induction of G_0_/G_1_ phase cell cycle arrest and apoptosis. In addition, apoptolidin A could inhibit the invasion and migration of CRC cells in vitro by suppressing the EMT process via the overexpression of NDRG1 in CRC cells. Therefore, the antimigration activity of apoptolidin A might be partly correlated with NDRG1 overexpression in CRC cells. Overall, this study provides new insights into the potential role of apoptolidin A in treating colorectal cancer. Moreover, apoptolidin A may be a promising therapeutic agent for the management of CRC metastasis.

## Figures and Tables

**Figure 1 pharmaceuticals-16-00491-f001:**
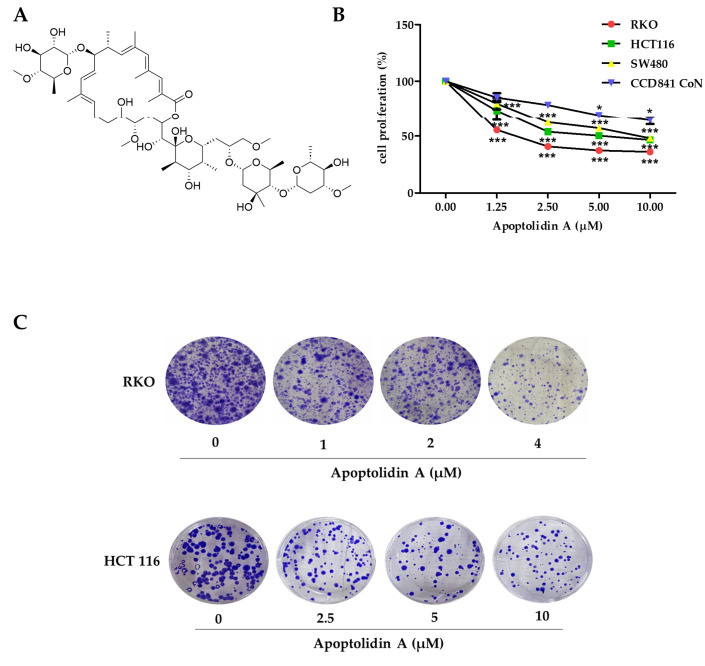
Effects of apoptolidin A on proliferation and colony formation in CRC cells. (**A**) The chemical structure of apoptolidin A. (**B**) Cells were seeded and treated with various concentrations of apoptolidin A for 72 h. The proliferative activity of RKO, HCT116, SW480, and CCD841 CoN (normal epithelial colon cell) cells were evaluated using the sulforhodamine B (SRB) assay. (**C**) The effect of apoptolidin A on colony formation. Cells were treated with apoptolidin A for 72 h, followed by additional incubation for 2 weeks. Data are presented as the mean values ± SD from three-independent experiments. * *p* < 0.05, *** *p* < 0.001 indicates statistically significant difference compared to the vehicle-treated control.

**Figure 2 pharmaceuticals-16-00491-f002:**
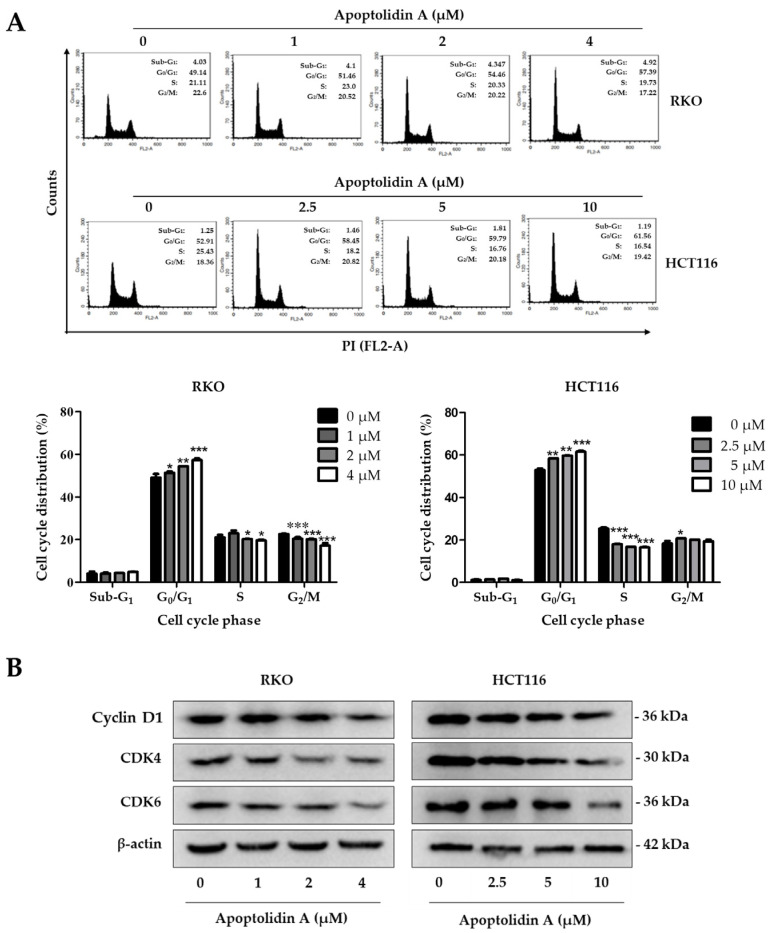
Effects of apoptolidin A on cell cycle regulation in CRC cells. (**A**) Induction of G_0_/G_1_ cell cycle arrest in RKO and HCT116 cells after 24 h treatment of apoptolidin A. The cell cycle distribution of RKO and HCT116 cells was analyzed by flow cytometry after staining with propidium iodide (PI). Data are presented as the mean values ± SD from three independent experiments. * *p* < 0.05, ** *p* < 0.01, and *** *p* < 0.001 indicate statistically significant difference compared with the vehicle-treated control. (**B**) Expression of cell cycle checkpoint proteins by Western blot analysis after 24 h treatment with apoptolidin A. β-Actin was used as an internal control.

**Figure 3 pharmaceuticals-16-00491-f003:**
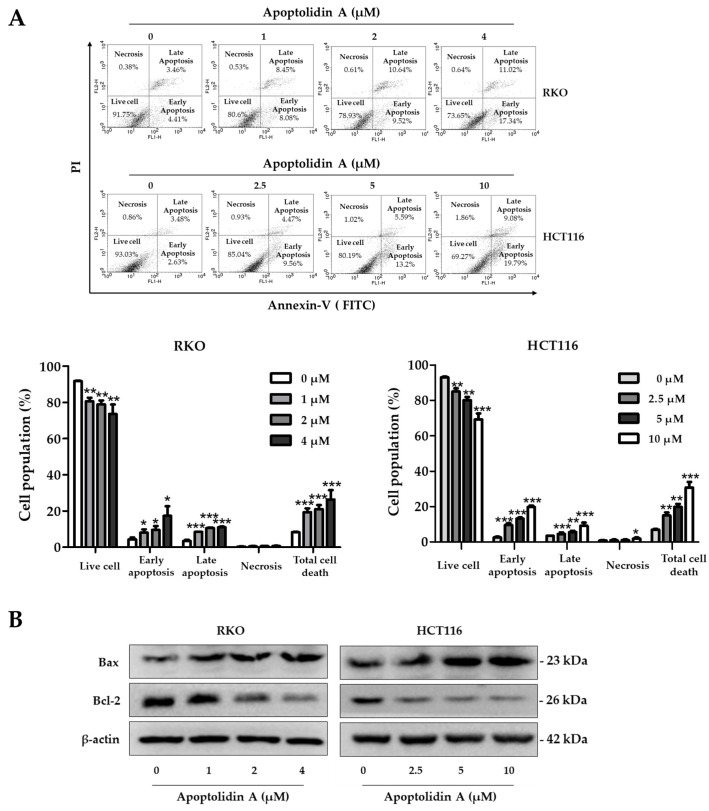
Effects of apoptolidin A on the induction of apoptosis in CRC cells. (**A**) RKO and HCT116 cells were treated with the indicated concentrations of apoptolidin A for 48 h before being stained with annexin V-fluorescein isothiocyanate (V-FITC) and PI for flow cytometry. Data are presented as the mean values ± SD from three-independent experiments. * *p* < 0.05, ** *p* < 0.01, and *** *p* < 0.001 indicate statistically significant difference compared to the vehicle-treated control. (**B**) Immunoblot analysis was used to assess the protein expression of Bax and Bcl-2 following a 48 h treatment of apoptolidin A. β-Actin was used as an internal control.

**Figure 4 pharmaceuticals-16-00491-f004:**
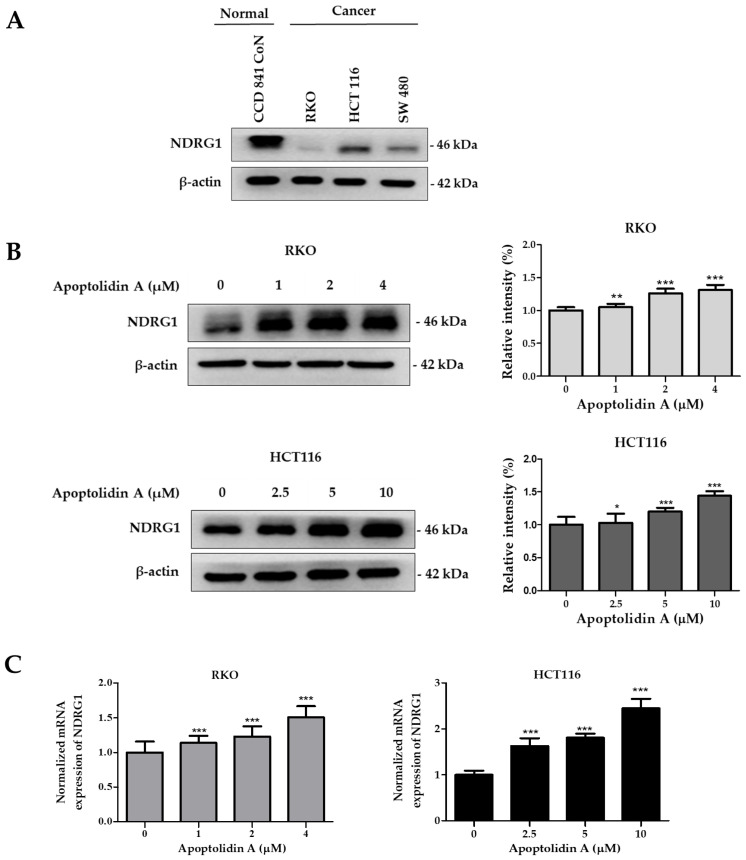
Effects of apoptolidin A on N-Myc downstream-regulated gene 1 (NDRG1) expression in CRC cells. (**A**) The expression of NDRG1 protein in normal epithelial colon cells and CRC cells was determined by Western blot analysis. β-Actin was used as an internal control. (**B**) Effects of apoptolidin A on NDRG1 protein expression after 24 h treatment in CRC cells. β-Actin was used as an internal control. (**C**) Real-time polymerase chain reaction (PCR) analysis of NDRG1 mRNA levels following 24 h treatment with apoptolidin A in CRC cells. Data are presented as the mean values ± SD from three independent experiments. * *p* < 0.05, ** *p* < 0.01, and *** *p* < 0.001 indicate statistically significant difference compared with the vehicle-treated control.

**Figure 5 pharmaceuticals-16-00491-f005:**
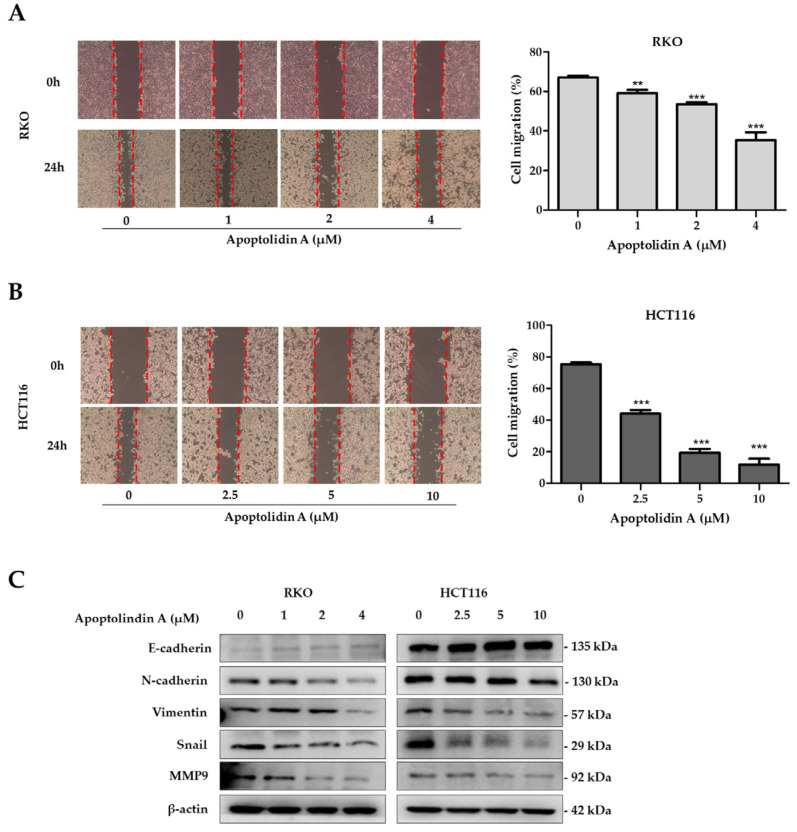
Effect of apoptolidin A on CRC cell migration. Wound healing assay of (**A**) RKO and (**B**) HCT116 cells after treatment with the indicated concentrations of apoptolidin A for 24 h. Images of a typical wound closure captured with a light microscope are displayed (left panel). Measurement of the wound area using Image J (right panel). Data are presented as the mean values ± SD from three independent experiments. ** *p* < 0.01, and *** *p* < 0.001 indicate statistically significant difference compared with the vehicle-treated control. (**C**) Immunoblotting was performed to assess the expression of E-cadherin, N-cadherin, vimentin, snail, and MMP9 proteins after 24 h treatment of apoptolidin A. β-Actin was used as an internal control.

**Figure 6 pharmaceuticals-16-00491-f006:**
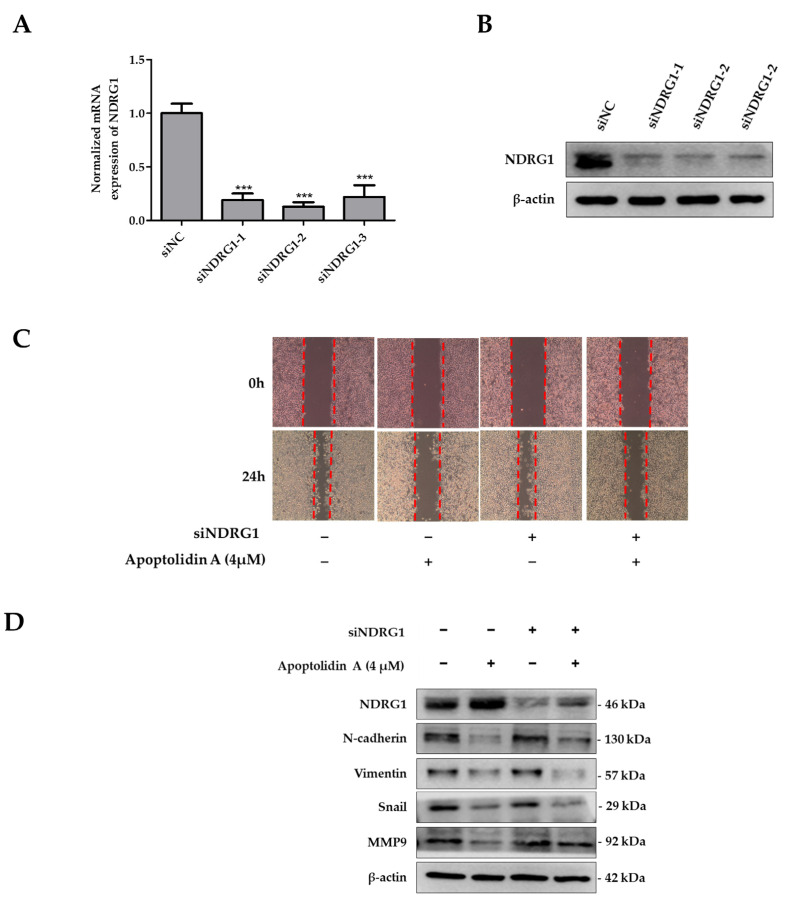
Effects of silencing NDRG1 on cell migration of apoptolidin A in RKO cells. The RKO cells were transfected with a set of NDRG1 siRNA for 48 h, and then PCR and immunoblot analysis were used to determine NDRG1 gene (**A**) and protein expression (**B**) in RKO cells. *** *p* < 0.001 indicate statistically significant difference compared with the siNC-treated control. (**C**) RKO cells were transfected with a selected siNDRG1 for 48 h and then seeded in six wells. A wound healing assay was used to assess the migration of RKO cells transfected with a particular siNDRG1 after 24 h apoptolidin A treatment. (**D**) RKO cells were transfected with siNDRG1 for 48 h and then treated with apoptolidin A for 24 h. The expression of EMT-related proteins was detected by immunoblotting. β-Actin was used as an internal control.

**Table 1 pharmaceuticals-16-00491-t001:** Antiproliferative activities of apoptolidin A in colorectal cancer cells.

Cell Lines	Apoptolidin A (IC_50_, μM)	Etoposide ^1^ (IC_50_, μM)
RKO HCT116	1.45 4.30	1.90 0.78
SW480 CCD841 CoN	8.57 >10	0.90 7.45

^1^ Etoposide was used as the positive control.

## Data Availability

Data are reported in the article.

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
