# Peer review of "Antimetastatic Activity of Apoptolidin A by Upregulation of N-Myc Downstream-Regulated Gene 1 Expression in Human Colorectal Cancer Cells"

_pharmaceuticals, 2023, doi:10.3390/ph16040491_

Round 1

Reviewer 1 Report

The authors in the manuscript characterize the effect of cyclic lactone apoptolidin on colorectal cell lines. They show a dose dependent growth inhibition of apoptolidin in three CRC cell lines, cell cycle arrest, migration, intrinsic apoptosis. Furthermore, they demonstrate that NDRG1 expression is restored following incubation with apoptolidin.  They further link the downregulation of NDRG1 by sirna leads to anti metastatic activity following apoptolidin exposure. Although restoration of NDRG1 is a desired feature, the mechanism around is vague. Authors should convince the reader on the specificity of NDRG1 in the treatment outcome. The readability is good and images are neat and good resolution.

Following issues need to be addressed before the manuscript is considered for publication.

Major Points

-        Authors do not describe how did they calculate the IC50

-        Cell proliferation assay and IC50 in a non-tumor cell line is required in Fig1

-        Authors should discuss the mechanism of NDRG1 elevation following apoptolidin exposure. Have they checked levels of other TSGs, eg. P53 etc

Minor Points

-        The WB images should be marked with their respective molecular weight in the figures

Reviewer 2 Report

The article is relevant research about investigation drug candidates that inhibit Colorectal cancer  metastasis.

The article demonstrate, that these findings suggest that apoptolidin A exerts antiproliferative and antimetastatic activities by regulating the NDRG1-activated EMT pathway in CRC cells. 

The English language and style should be checked. In the conclusions, the importance of this study should be emphasized more.

The statistical value t and p should be in Italic (in all article).

The article is accepted to publication.

Round 2

Reviewer 1 Report

The Authors have addressed the comments appropriately.

Manuscript is recommended for publication.